# Genetic Associations between Voltage-Gated Calcium Channels and Psychiatric Disorders

**DOI:** 10.3390/ijms20143537

**Published:** 2019-07-19

**Authors:** Arturo Andrade, Ashton Brennecke, Shayna Mallat, Julian Brown, Juan Gomez-Rivadeneira, Natalie Czepiel, Laura Londrigan

**Affiliations:** Department of Biological Sciences, University of New Hampshire, Durham, NH 03824, USA

**Keywords:** voltage-gated calcium channels, major depressive disorder, autism spectrum disorder, schizophrenia, bipolar disorder, attention-deficit and hyperactivity disorder, anxiety, calcium channel modulators, psychiatric disorders, auxiliary subunits, genetic risk variations

## Abstract

Psychiatric disorders are mental, behavioral or emotional disorders. These conditions are prevalent, one in four adults suffer from any type of psychiatric disorders world-wide. It has always been observed that psychiatric disorders have a genetic component, however, new methods to sequence full genomes of large cohorts have identified with high precision genetic risk loci for these conditions. Psychiatric disorders include, but are not limited to, bipolar disorder, schizophrenia, autism spectrum disorder, anxiety disorders, major depressive disorder, and attention-deficit and hyperactivity disorder. Several risk loci for psychiatric disorders fall within genes that encode for voltage-gated calcium channels (Ca_V_s). Calcium entering through Ca_V_s is crucial for multiple neuronal processes. In this review, we will summarize recent findings that link Ca_V_s and their auxiliary subunits to psychiatric disorders. First, we will provide a general overview of Ca_V_s structure, classification, function, expression and pharmacology. Next, we will summarize tools to study risk loci associated with psychiatric disorders. We will examine functional studies of risk variations in Ca_V_ genes when available. Finally, we will review pharmacological evidence of the use of Ca_V_ modulators to treat psychiatric disorders. Our review will be of interest for those studying pathophysiological aspects of Ca_V_s.

## 1. Introduction

Voltage-gated calcium channels (Ca_V_s) are transmembrane protein activated by depolarization of membrane potential. The calcium that enters through Ca_V_s is crucial for cellular processes including gene expression, hormone release, neurotransmitter release, cardiac muscle contraction, and pacemaker activity [1]. Based on their activation threshold, Ca_V_s are classified as high or low voltage activated (HVA and LVA). HVA Ca_V_s form multi-protein complexes comprised of the Ca_V_α_1_ pore-forming and the auxiliary subunits, Ca_V_α_2_δ and Ca_V_β (Table 1). These auxiliary subunits have profound effects on the biophysical properties and membrane targeting of the Ca_V_α_1_ subunit [2,3]. Targeted deletions or disruptive mutations of genes encoding Ca_V_α_1_, Ca_V_α_2_δ and Ca_V_β subunits result in deleterious effects, highlighting the importance of these genes [2,3,4,5,6,7,8]. Classically, dysfunction of Ca_V_s has been linked to neurological disorders including Parkinson’s disease, epilepsy, migraine, ataxia and neuropathic pain [9,10,11,12,13,14,15]. More recently, due to the advancement in genetic techniques to sequence and analyze full human genomes, genes encoding Ca_V_s have been linked to psychiatric disorders [1,14,16]. All of this combined expands the relevance of Ca_V_s in health and disease.

Ca_V_s are being considered as molecular targets to treat several neurological conditions including psychiatric disorders [14]. Furthermore, functional studies of Ca_V_ gene risk variations identified in patients with psychiatric disorders are providing mechanistic insights into these conditions. In this review, we will summarize literature on the structure and function of Ca_V_ genes, we will briefly overview some of the genetic tools that have allowed researchers to establish genetic links between Ca_V_s and psychiatric disorders, then we will examine studies that have linked Ca_V_ genes to several psychiatric disorders including bipolar disorder (BD), schizophrenia (SCZ), autism spectrum disorders (ASD), anxiety disorders, major depressive disorder (MDD), and attention-deficit and hyperactivity disorder (ADHD). If available, we will provide a summary of behavioral studies performed in animal models where Ca_V_ function has been disrupted and a summary of functional studies of risk variations in Ca_V_ genes identified in genetic screenings. Here we will only review phenotypes that are related to psychiatric disorders; for an in-depth analysis of animal models with targeted disruption of Ca_V_ genes see the following reviews [14,17,18,19,20,21,22,23,24,25]. Finally, we will summarize literature on therapeutic strategies that focus on Ca_V_s as pharmacological targets. In this review, we will utilize the gene name for Ca_V_s when referring to variations in the gene, and the protein name for Ca_V_s when referring to the channel (Table 1).

## 2. Structure and Pharmacology of Voltage-Gated Calcium Channels

### 2.1. Ca_V_α_1_ Subunits

Ten genes encode the Ca_V_α_1_-pore-forming subunit of Ca_V_s (*CACNA1*). Based on their pharmacology and sequence similarity, Ca_V_α_1_ subunits are subdivided in three subfamilies (Ca_V_1, Ca_V_2 and Ca_V_3) (Table 1). In this review, we will briefly summarize pharmacological aspects of Ca_V_s, for those interested in a more comprehensive analysis of pharmacological agents that target Ca_V_s; please see [25,26,27,28,29]. The Ca_V_1 channel subfamily is comprised of Ca_V_1.1 (*CACNA1S*), Cav1.2 (*CACNA11C*), Ca_V_1.3 (*CACNA1D*) and Ca_V_1.4 (*CACNA1F*) channels. Ca_V_1 channels are sensitive to dihydropyridines (DHPs) and exhibit long-lasting activity relative to the members of Ca_V_2 and Ca_V_3, hence these channels are also known as L-type [28]. 

The Ca_V_2 channel subfamily is comprised of Ca_V_2.1 (*CACNA1A*), Ca_V_2.2 (*CACNA1B*), and Ca_V_2.3 (*CACNA1E*). Ca_V_2.1, Ca_V_2.2, and Ca_V_2.3 generate the P/Q-type, N-type and R-type currents respectively. These channels are generally localized in presynaptic terminals where they control calcium-dependent transmitter release in central and peripheral synapses, although Ca_V_2.3 is also present in dendrites and extra postsynaptic sites [28]. Ca_V_2 channels are selectively blocked with toxins. Ca_V_2.1 is sensitive to ω-agatoxin IVA, Ca_V_2.2 to ω-conotoxin GVIA, and Ca_V_2.3 to the SNX-482 peptide toxin [28]. 

The Ca_V_3 subfamily is comprised of Ca_V_3.1 (*CACNA1G*), Ca_V_3.2 (*CACNA1H*), and Ca_V_3.3 (*CACNA1I*), which generate T-type currents. Ca_V_3 channels exhibit small single channel conductance, and relatively lower threshold of activation compared to all members of the Ca_V_1 and Ca_V_2 subfamilies [27]. It is important to note that Ca_V_1.3 channels exhibit a threshold of activation that is lower relative to the other members of the Ca_V_1 subfamily and Ca_V_2 channels, but slightly higher than all the Ca_V_3 subfamily members [30]. All the *CACNA1* genes undergo extensive alternative splicing that produces various splice variants with differences in their tissue expression, pharmacology and biophysical properties [31]. In some cases, the pharmacological and biophysical properties of a given splice variant overlap with those ones from members of a different Ca_V_ subfamily [32]. 

The primary structure of the Ca_V_α_1_ pore-forming subunit is organized into four homologous domains (DI-IV). Each domain contains six membrane-spanning segments (S1–S6), with a re-entrant loop between S5 and S6, which contains negatively charged residues (glutamates and/or aspartates) that are essential for the selectivity filter. S4 contains positively charged residues (arginines) that function as voltage-sensors. The amino and carboxyl termini, as well as linker sequences between the DI-II, DII-III, and DIII-IV are all cytosolic. These sites are important for the interaction of Ca_V_α_1_ with intracellular proteins, as well as signaling cascades that regulate calcium entry through Ca_V_s (Figure 1) [33]. 

### 2.2. Ca_V_ Auxiliary Subunits, Ca_V_α_2_δ and Ca_V_β

Members of the Ca_V_1 and Ca_V_2 subfamilies form membrane complexes with the auxiliary subunits Ca_V_α_2_δ and Ca_V_β, influencing several biophysical and pharmacological properties of the Ca_V_α_1_ subunit [2,3,22]. 

#### 2.2.1. Ca_V_α_2_δ Subunits 

Four genes exist for the Ca_V_α_2_δ subunits (*CACNA2D1-4*), which encode Ca_V_α_2_δ-1-4 proteins [22]. Each Ca_V_α_2_δ subunit is translated from a single gene, which produces a protein that is cleaved into the α_2_ and δ peptides. A disulfide bond links these peptides [34,35]. Ca_V_α_2_δ is entirely extracellular, but it is attached to the cell membrane by a glycosylphosphatidylinositol anchor (GPI) domain (Figure 1) [36]. Interestingly, the α_2_ peptide contains structural domains such as the von Willebrand factor A domain (vWA) and two Cache domains [37]. The vWA domain in Ca_V_α_2_δ contains a metal-ion-adhesion site (MIDAS) that is important for membrane trafficking [38]. The functional role of the Cache domains is poorly understood [18,20]. Ca_V_α_2_δ-1 and Ca_V_α_2_δ-2 are targets for the gabapentinoid drugs, gabapentin and pregabalin [39]. Similar to *CACNA1* genes, *CACNA2D* genes are also subject to extensive alternative splicing that impacts affinity for gabapentinoid drugs and other functions of the Ca_V_α_2_δ subunits [40,41].

#### 2.2.2. Ca_V_β Subunits 

Four genes exist for the Ca_V_β subunits (*CACNB1-4*), which encode Ca_V_β_1_–Ca_V_β_4_. Ca_V_β subunits are located in the cytoplasm (Figure 1); however, some splice variants of Ca_V_β_2_ are attached to the membrane via a palmitoylation site [42,43], and both Ca_V_β_3_ and the splice variant Ca_V_β_4c_ can be mobilized to the nucleus [44,45,46]. Ca_V_β subunits contain three conserved domains: an inactive guanylate kinase domain (GK), an *src* homology domain 3 (SH3), and a HOOK region [47,48,49,50]. The Ca_V_β-GK domain is important for the interaction with the AID domain in the I-II loop of the Ca_V_α_1_ subunit [50]. The Ca_V_β-SH3 and HOOK domains mediate specific protein–protein interactions of Ca_V_β subunits, for example, with dynamin [51]. All *CACNB* genes undergo alternative splicing [52].

## 3. General Function of Voltage-Gated Calcium Channels and Auxiliary Subunits

### 3.1. Ca_V_α_1_ Subunits 

Ca_V_ channels are expressed in a wide variety of tissues where they serve specific functions. Ca_V_1.1 is restricted to the skeletal muscle where the movement of the gating mechanisms induced by depolarization leads to opening of the ryanodine receptors (RYR), a class of calcium channels located in the sarcoplasmic membrane. The opening of RYR increases intracellular calcium, which results in activation of calcium-dependent contractile proteins [53]. Ca_V_1.2 and Ca_V_1.3 are broadly co-expressed in various tissues including the brain, heart, smooth muscle, and neurosecretory systems. These two channels are important for gene expression, calcium transients in dendrites, and the coupling of electrical signals to hormone secretion [54,55,56,57,58]. Ca_V_1.2 controls contraction of heart muscle, and together with Ca_V_1.3 controls the pacemaking activity of midbrain dopaminergic neurons and adrenal chromaffin cells [59,60,61,62]. Ca_V_1.3 is key for the pacemaking firing of the sinoatrial node and for transmitter release from hair cells of the inner ear [63,64,65]. Ca_V_1.4 controls glutamate release from photoreceptors [66,67]. 

Ca_V_2.1, Ca_V_2.2, and Ca_V_2.3 are involved in the release of neurotransmitters. Ca_V_2.1 and Ca_V_2.2 channels have a dominant role in the release of fast transmitters such as GABA, acetylcholine, and glutamate [68,69]. Ca_V_2.2 channels are dominant in peripheral terminals that release glutamate and noradrenaline, as well as in central synapses that release dopamine, serotonin and noradrenaline [70,71,72]. Ca_V_2.2 channels are also dominant in interneurons that express the cholecystokinin peptide [73,74]. Ca_V_2.3 channels are present in the presynaptic terminals and dendrites of certain synapses of the central nervous system [75]. In the presynaptic terminals, Ca_V_2.3 channels are localized in the active zones or in their periphery thereby controlling transmitter release [76]. In the dendrites, Ca_V_2.3 channels control calcium-dependent spikes [77]. G-protein coupled receptors for several neurotransmitters including GABA, endogenous opioids, and endocannabinoids heavily regulate Ca_V_2 channels [78,79]. This is an important negative feedback mechanism to limit the release of neurotransmitter [78,79]. Ca_V_2 channels interact with soluble N-ethylmaleimide-sensitive fusion protein receptors (SNAREs), which promote the fusion of secretory vesicles to the membrane in a calcium-dependent manner. This calcium generally enters through Ca_V_2 channels [80].

Ca_V_3.1, Ca_V_3.2, and Ca_V_3.3 channels control repetitive firing and pacemaking activity [27]. Ca_V_3 channels open at relatively low voltages compared to members of the Ca_V_1 and Ca_V_2 subfamilies and have fast voltage-dependent inactivation. These unique biophysical properties underlie the role of Ca_V_3 channels in rhythmic firing of action potentials [25,81]. Ca_V_3 channels control the pacemaking activity of the sinoatrial node in the heart [82,83], and the rhythmic bursts of action potential in relay neurons in the thalamus [84]. Ca_V_3 channels are not known to be associated with the auxiliary subunits Ca_V_α_2_δ and Ca_V_β; however, recent evidence suggests that Ca_V_3 channels interact with CACHD1, a protein closely related to the Ca_V_α_2_δ subunits (Table 1) [19,85,86].

### 3.2. Ca_V_α_2_δ Subunits 

Ca_V_α_2_δ-1 is expressed in skeletal, cardiac and smooth muscle, secretory systems; central and peripheral neurons [39]. Ca_V_α_2_δ-2 is abundantly expressed in the cerebellum, and to a lesser extent in other areas of the brain [8]. Ca_V_α_2_δ-3 is expressed throughout the central and peripheral nervous systems [21,87]. Finally, Ca_V_α_2_δ-4 expression is limited to the retina and endocrine tissue [88]. Expression of Ca_V_α_2_δ subunits increases membrane trafficking, stabilizes Ca_V_s complexes in the cell surface and produces shifts in voltage-dependence of activation as well as inactivation [3,20]. Ca_V_α_2_δ subunits promote synaptogenesis by binding to thrombospondin [89], influence neurotransmission through interaction with α-neurexins [90,91], and affect synaptic plasticity by interacting with N-methyl-D-aspartate (NMDA) receptors [92]. 

### 3.3. Ca_V_β Subunits 

Ca_V_β subunits are broadly expressed in several tissues including brain, heart and skeletal muscle. These proteins promote trafficking of Ca_V_α_1_ to the cell surface by occluding endoplasmic reticulum (ER) retention signals present in the linker between DI and DII of Ca_V_α_1_ [93]. Ca_V_β subunits are key for modulation of Ca_V_1 and Ca_V_2 channels by G-protein coupled receptors and other signaling complexes including Ras-related GTPases [17,78]. Ca_V_β3 and particularly Ca_V_β4 are thought to induce gene expression [44,46]. 

## 4. Genetic Analysis and Tools to Study Psychiatric Disorders

Few cases exist where the inheritance of a disorder involving Ca_V_ genes follows mendelian models. However, spinal cerebellar ataxia 6 (SCA6) and Timothy syndrome (TS) are two cases that follow an autosomal dominant pattern. Alterations in the *CACNA1A* and *CACNA1C* genes underlie SCA6 and TS, respectively [94,95]. In SCA6, the *CACNA1A* gene contains between 20 and 33 CAG repeats that encode glutamines in the C-terminus [96]. Although the molecular mechanism by which these repeats lead to the disease remains to be fully understood, current evidence suggests transcriptional dysregulation mediated by a Ca_V_2.1 C-terminus fragment with the glutamine repeats [97]. In TS, mutations in *CACNA1C* (G402S and G406R) produce Ca_V_1.2 channels with gain of function, and these mutations are located in the C-terminus of DIS6 [98,99]. TS is a condition that affects the heart and the nervous system. Patients with Timothy syndrome present several characteristics seen in patients with ASD [95]. Recent evidence suggests that Ca_V_1.2 mutations underlying TS produce defects in neuronal migration during cortical development [100]. In contrast to SCA6 and TS, most psychiatric disorders are genetically complex conditions that involve the interaction among several genes and their interactions with the environment [101,102]. 

### 4.1. Genetic Strategies to Study Psychiatric Disorders 

Several genetic methods have been used to determine the genes or set of genes that are likely to underlie psychiatric disorders. Historically, linkage and linkage-disequilibrium studies provided the initial evidence of the genetic origins of psychiatric disorders [103,104]. However, it is now possible to perform genetic analysis using whole genomes from large populations through genome-wide association studies (GWAS) to discover new risk variations associated with psychiatric disorders [105]. 

#### 4.1.1. Linkage Studies

Evidence for linkage is derived from observing the cosegregation of specific genomic regions with a given disorder. As such, this method is most effective for the study of disorders inherited in a Mendelian fashion. Pedigrees containing multiple generations of genetic data can be used to elucidate inheritance patterns and map potential genomic risk loci for a given disorder. The identification of large families with high prevalence of a given condition often facilitates linkage studies. In these studies, the inheritance of a genetic loci can be correlated with the presence or absence of the disorder [103]. 

#### 4.1.2. Linkage-Disequilibrium Studies

In these studies, the aim is to map a nonrandom association of alleles at two or more loci to discover disease haplotypes. These haplotypes are thought to be inherited from one or a few founding members of isolated populations [106]. 

#### 4.1.3. Association Studies 

Here the goal is to find risk loci for a specific condition by assessing correlations between disease status and genetic variation. Of the association studies, GWAS are becoming a popular method to screen genetic variations of disease across whole genomes of large populations. GWAS have identified several genetic variations of Ca_V_ genes linked to BD, SCZ, ASD, ADHD, and MDD [14,16]. We will review several of these cases below. Thanks to GWAS data, many new risk loci for psychiatric disorders have been found [107,108]. 

All the genetic approaches mentioned above have helped to identify associations of several gene variations to psychiatric disorders. These variations include single nucleotide polymorphisms (SNPs), small indels, copy number variations (CNVs), *de novo* variations, and large chromosomal rearrangements [109]. 

### 4.2. Tools to Identify and Analyze Genetic Variations Associated with Psychiatric Disorders

Our understanding of psychiatric disorders is evolving towards a more comprehensive analysis that includes genetic, genomic, functional and behavioral studies. These are possible thanks to tools that facilitated the screening of large cohorts of patients (probands) and their corresponding unaffected relatives. Among these tools are next-generation sequencing, microarrays, endophenotype analysis, gene network analysis and computational modeling [109]. 

#### 4.2.1. Next Generation Sequencing (NGS) 

These tools include whole exome and genome sequencing (WES and WGS, respectively), as well as RNA sequencing (RNA-seq). WES detects genetic variations through capture and sequencing of coding regions within the genomic DNA. Since most of the genome is noncoding, this approach greatly reduces the amount of sequencing to ~2% of the whole genome [110]. WGS offers an almost complete sequence coverage (~95%) that includes coding and non-coding regions and is more powerful to detect exome variations than WES. This increased coverage enables identification of non-coding regions that include splicing regulatory elements, promoters, enhancers and sites that regulate RNA transport and stability [111]. WES are more commonly used in genetic screenings for psychiatric disorders because of their lower cost compared to WGS [112]. Nevertheless, WES studies allow us to focus on regions where variations can be identified and interpreted faster than in WGS studies [112]. 

RNA-seq is a common tool used for genetic analysis of psychiatric disorders. This technology enables quantification of gene expression, detection and quantification of exon splicing, quantification of rare transcripts and non-coding RNAs, and detection of genome rearrangements. In summary, RNA-seq provides a whole transcriptome landscape with high signal to noise ratio and with small amount of RNA input [113].

#### 4.2.2. Microarrays

Studies using microarrays are commonly used to identify genetic risk variations that involve structural changes >1000 bp [109,114]. Large structural variations detected by microarrays are thought to increase the risk SCZ, ASD, and ADHD [115,116,117,118,119,120,121]. 

#### 4.2.3. Gene Network Analysis 

The discovery of risk variations associated with psychiatric disorders has been a stepping stone to elucidate the molecular mechanisms that underlie these conditions. Now the challenge is integrating this information to understand how genetic variations influence complex disorders and traits [122,123]. It is thought that the complex interactions of genes in a network are more likely to explain phenotypes of psychiatric disorders, rather than the additive effect of those genes. Complex interactions of genes within networks include transcriptional regulatory, protein–protein interaction, metabolomic networks, and a hierarchical interaction with other gene networks [124,125,126,127]. Furthermore, complex interactions between gene networks with the environment are becoming increasingly important to fully explain phenotypes linked to psychiatric disorders [128,129,130,131,132,133]. 

#### 4.2.4. Endophenotypes 

Despite recent advances linking genetic risk variations to psychiatric disorders, the phenotypic consequences of those variations are poorly understood. However, a combination of molecular genetics with endophenotypes might represent a promising approach to understand the behavioral links between risk variations and psychiatric disorders [134]. Endophenotypes are quantitative neurobehavioral traits that are associated with a disorder, are reasonably heritable, co-segregate with the disease and are independent of the clinical status of the disorder [135]. Endophenotypes provide clinical measures of disease diagnosis and progression. Examples of endophenotypes include deficits in pre-pulse inhibition and sensory gating, decline in working memory, and deficits in face emotion labeling [134,135]. Interestingly, the latter has been associated with *CACNA1C* in patients with bipolar disorder [136].

#### 4.2.5. Computational Psychiatry 

Mathematical approaches are being used to integrate findings derived from genetic screenings, functional studies of gene risk variations, and behavioral phenotypes. Computational psychiatry is an emerging field that aims to model the compounded effects of individual genes, as well as their interaction with other genes (gene networks) and with the environment using mathematics [137,138,139,140]. Computational approaches have been successfully used to provide insightful mechanisms for disorders such as SCZ, ASD, and ADHD [141,142,143]. 

## 5. Genetic Associations between Ca_V_ Genes and Psychiatric Disorders

Gene network analyses have consistently implicated Ca_V_ genes in psychiatric disorders, which nicely correlates with the role calcium signaling in neuronal function [14,16]. In this section, we will review several large studies that have provided strong evidence linking Ca_V_ genes to psychiatric disorders and related endophenotypes. We will also briefly describe functional studies, when available, of risk variations for Ca_V_ genes.

### 5.1. Ca_V_1.2

Ca_V_1.2 channels are ion channels that have been extensively studied in the heart. Here, Ca_V_1.2 channels tightly couple depolarization to muscle contraction through activation of RYR located in the ER of cardiomyocytes [144]. Additional studies have demonstrated that Ca_V_1.2 is expressed in postsynaptic terminals in the brain, and together with Ca_V_1.3, influences neuronal firing and couples excitation to gene expression [145]. The activity of neuronal Ca_V_1.2 and Ca_V_1.3 channels is implicated in several processes relevant to psychiatric disorders including learning, memory, and brain development [146,147]. Although global disruption of Ca_V_1.2 channels leads to embryonic lethality in mouse [148], studies of conditional knock out (KO) mice have shown that Ca_V_1.2 channels are involved in high-order brain functions such as spatial memory and remote spatial memory consolidation [149,150]. Furthermore, Ca_V_1.2 KO heterozygous mice exhibit increased anxiety-like behavior [151,152]. In line with this, deletion of Ca_V_1.2 in calcium/calmodulin dependent-protein kinase IIα expressing cells (mostly forebrain neurons) also leads to anxiety-like behavior [153]. Alterations in fear is a trait widely observed in patients with anxiety disorders, and conditional Ca_V_1.2 KO mouse models have shown alterations in fear responses. Functional ablation of Ca_V_1.2 in Nestin expressing cells leads to reduced acquisition of conditioned fear [154], and specific deletion of Ca_V_1.2 in the anterior cingulate cortex impairs observational fear [155]. Mice harboring loss of Ca_V_1.2 channels in glutamatergic neurons of the forebrain show social behavior deficits associated with the prefrontal cortex [153]. All this evidence provides strong support for the involvement of Ca_V_1.2 channels in psychiatric disorders. Risk variations in the gene that encodes for Ca_V_1.2, *CACNA1C*, have been found in several association studies of BD, and evidence suggest that some of *CACNA1C* variations are risk for SCZ, MDD, ADHD, and ASD. 

Several SNPs in *CACNA1C* have been linked to psychiatric disorders with most of them being located in a large intron (~330 kb) between exons 3 and 4 (intron 3). Significant association of the SNP rs1006737 allele to BD was originally found in a European cohort (>4300 cases and >6000 controls) [156]. Associations of this SNP with BD have been replicated in several other studies; furthermore significant association of rs100637 with SCZ, ADHD and MDD has also been detected [157,158,159]. At the molecular level, rs100637 is correlated with changes in *CACNA1C* expression, including decreased expression in the cerebellum [160], but increased expression in the dorsolateral prefrontal cortex and induced human neurons [136,161]. The latter observation correlates with increased L-type currents seen in induced human neurons derived from individuals carrying rs1006737 [161]. Furthermore, the minor allele for rs1006737 (A) is associated with increased methylation of CpG islands located within intron 3 [162]. Imaging studies have shown associations of rs1006737 with changes in structure and activity of brain regions related to emotion processing, memory formation and cognition, including hippocampus, inferior occipital fusiform gyrus, prefrontal cortex and amygdala [136,163,164]. For example, carriers of rs1006737 show greater thickness of the medial orbitofrontal cortex than non-carriers, and the presence of this SNP correlates with age-related caudal anterior cingulate cortex thickening [164]. In addition, two independent studies have shown that rs1006737 is associated with increased amygdala volumes in adults and adolescents [165,166]. Behavioral studies in humans suggest that rs1006737 is linked to facial emotion recognition in both healthy individuals and patients with BD [167,168]. Some studies suggest that rs1006737 is also associated with borderline personality disorder in females, but not males [169,170]. Furthermore, rs1006737 has significant association with reduced baseline affective startle modulation in healthy males. Alterations in this endophenotype have been observed in severely depressed and anxious patients, as well as patients with BD in remission [171].

As mentioned above, rs1006737 has also shown strong associations with SCZ. Additive interaction of the SNP rs1006737 *CACNA1C* with rs1344706 in the zinc finger protein 804A gene (ZNF804A) has been linked to defects in white matter microstructure and psychosis [172], although the effect of rs1344706 is thought to be larger than rs1006737. In MDD, rs1006737 was associated with less baseline depressive severity [173]. Furthermore, rs1006737 showed biphasic association with antidepressant treatment in a European population. The A allele was associated with a better outcome of antidepressant treatment, but it shows the opposite association in a group of individuals with treatment-resistant depression [174]. 

The SNP rs2007044 has been associated with SCZ in several studies including Asian, East Asian, European and Ashkenazi Jew populations [175,176,177,178,179]. This SNP was associated with decreased functional connectivity between the right dorsolateral prefrontal cortex and right superior occipital gyrus/cuneus, as well as the anterior cingulate cortex; and at the behavioral level with poor working memory performance [180]. Also, rs2007044 is associated with increased concentrations of glutamate, glutamine and glutamate, plus glutamine in subcortical regions such as basal ganglia and thalamus. These observations have been reported in patients with SCZ, especially in subjects at risk of psychosis [181].

Sleep disturbance is consistently reported in patients with psychiatric disorders including SCZ, BD and MDD [182,183,184]. *CACNA1C* variations in intron 3 have been linked to sleep traits such as narcolepsy (rs10774044), sleep latency and sleep quality (rs7316184, rs7304986, rs7301906, rs16929275, rs16929276, rs16929278, rs2051990) [185,186,187]. The SCZ risk variations in *CACNA1C* (rs4765913, rs4765914, and rs2239063) are associated with sleep latency in infants [188]. Allele rs4765914, together with rs7297582, was identified in two independent studies as genetic risks for BD, MDD, and SCZ [158,189]. 

Other SNPs in *CACNA1C* have been linked to several psychiatric conditions. The SNP rs73248708 (intron 3) and rs116625684 (intron 1) were not associated with SCZ or other psychiatric disorders, but they affect the risk of developing depressive symptoms upon exposure to adult severe trauma in adulthood [190]. The SNP rs10848635 was identified in a Korean and in a Taiwanese population as risk factor for SCZ and BD, respectively [191,192]. Associations of rs10848635 with efficacy of the anti-depressant citalopram were also found [174]. The SNP rs4765913 was identified in two independent GWAS of European cohorts as genetic risk for BD [193,194]. The alleles rs10848653 and rs2239118 were identified using a family-based association test (parent/affected child trios) and linked to ASD, this study also identified SNPs in *CACNA1G* (see below) [193,194]. 

In addition to variations with mendelian inheritances (TS) and SNPs associated with psychiatric disorders, two *de novo* missense variations in *CACNA1C* were identified in a large whole-exome sequencing study using massively parallel short-read sequences of more than 2500 patients with SCZ and more than 2500 control subjects in a Swedish population [195]. One risk variant (G/T) is predicted to alter a canonical splice donor site for exon 21, which is part of a pair of mutually exclusive exons (together with exon 22), with exon 21 being dominant in the brain [31]. Exons 21 and 22 encode part of the extracellular loop between DIIS1 and DIIS2 and a portion of DIIS2 in Ca_V_1.2. The second risk variant (C/T) introduces a premature stop codon in the intracellular linker between DIII and DIV [16]. 

### 5.2. Ca_V_1.3

The *CACNA1D* gene encodes Ca_V_1.3. This channel contributes to the rhythmic activity of the sinoatrial node and thereby involved in the regulation of heart rate [145]. As stated above, Ca_V_1.3 shares some functions with Ca_V_1.2 in the brain. However, Ca_V_1.3 is the main contributor to the pacemaking activity of dopaminergic neurons in the substantia nigra [62]. Mouse genetic models of Ca_V_1.3 have provided information on the potential role of Ca_V_1.3 in psychiatric disorders. Ca_V_1.3 KO mice show anxiety-like phenotypes [196], although recent evidence suggests that these phenotypes are related to hearing deficits [197,198]. However, in an animal model where Ca_V_1.2 was mutated to confer resistance to DHPs (Ca_V_1.2DHP^−/−^), thereby allowing specific pharmacological manipulation of Ca_V_1.3 channels, activation of these channels led to depressive-like behaviors [198,199,200]. Finally, strong evidence suggests that Ca_V_1.3 channels play a key role in drug seeking behavior, a behavioral trait linked to addiction [147]. Interestingly, addiction is often found as a comorbidity with psychiatric disorders including BD and SCZ.

Risk variations in *CACNA1D* have been associated with BD, SCZ, ADHD, MDD, and ASD. The non-coding SNP rs893363, located in the 3′ UTR of *CACNA1D* and the putative promoter region of the choline dehydrogenase gene (*CHDH*), was found in a genome-wide analysis of five major psychiatric disorders including BD, SCZ, ADHD, MDD, and ASD [158]. In a study with samples from a cohort of European-American individuals, 111 non-coding variations in regulatory elements that are predicted to modify binding of transcription factors to genomic regions of *CACNA1D* show significant association with BD [201]. Furthermore, two coding variations in *CACNA1D* (A1751P and R1771W) located in the C-terminus segregate with BD type I cases in a large pedigree [202]. Although a study in a Han Chinese population found no association between *CACNA1D* SNPs and SCZ [203], more recent studies that include larger populations of East Asian, Chinese, European and Ashkenazi Jewish individuals identified the SNP rs2358740 located in a putative promoter region for *CACNA1D* and the mRNA decapping enzyme 1A gene (*DCP1A*) as a risk variant for SCZ [176,204,205]. 

Several studies point to links between *CACNA1D* and ASD. Through whole-exome sequencing three *de novo* missense variations in the linker between DI and DII in Ca_V_1.3 (A749G, G407R, and V401L) were identified as genetic risks for patients with sporadic autism and intellectual disability [206,207,208]. These genetic risk variations produce a gain of function of Ca_V_1.3 channels [209,210,211]. Additional variations (A59V, S1977L and R2021H) were also identified using WES. The A59V maps to an N-terminal region of Ca_V_1.3 that is key for calcium-dependent inactivation. S1977L and R2021H map to a proline-rich domain of the C-terminus that interacts with SH3 and Multiple Ankyrin Repeat Domain 3 protein (Shank3) [212]. Interestingly, *SHANK3* is another gene strongly linked to ASD [213,214]. The gene *CACNA1D* is subject to alternative splicing. Interestingly alterations in the relative abundance of several alternatively spliced exons in *CACNA1D* have been observed cortical samples of patients with ASD [215]. Finally, the variation Q567H is located between S1 and S2 of DII in Ca_V_1.3 and is linked to moderate hearing impairment and intellectual disability. This variation results in a loss of function [216]. 

Although risk variations in *CACNA1S* and *CACNA1F* encoding Ca_V_1.1 and Ca_V_1.4 channels have been identified in GWAS and WES studies for BD and SCZ, we will not review them here because the expression of these two genes in the brain is extremely rare relative to the other Ca_V_ genes; therefore the links between their corresponding risk variations and psychiatric disorders are hard to infer [16]. 

### 5.3. Ca_V_2.1

*CACNA1A* encodes Ca_V_2.1 channel, which is the most dominant presynaptic calcium channel in central synapses, particularly those ones from Purkinje cells in the cerebellum and excitatory synapses of cortex and hippocampus. Although Ca_V_2.1 KO mice are postnatally lethal [217,218,219], forebrain ablation of Ca_V_2.1 channels results in deficits in spatial learning and memory, and increased exploratory behavior suggesting a potential role of this channel in psychiatric-related phenotypes [220]. Various mutations in *CACNA1A,* causing gain or loss of function, have been found in patients with hemiplegic migraine 1 (FHM-1), Episodic Ataxia 2 (EA-2), SCA-6, and epilepsy [10]. More recently, a clinical recharacterization of patients with EA-2 and SCA-6 showed that they also present delayed development, endophenotypes related to learning disabilities, ASD and ADHD [221]. In another study, some FHM-1 and EA-2 patients also presented SCZ, learning disabilities and ADHD [222]. Furthermore, analysis of the splice isoform landscape across several psychiatric disorders show that alternative splicing of *CACNA1A* is altered in ASD [223]. Finally, rs10409541 was among the top 15 most contributory SNPs for ASD diagnosis prediction in a Central European population [194]. 

### 5.4. Ca_V_2.2 

*CACNA1B* encodes Ca_V_2.2 channels, which are dominant in presynaptic terminals of dorsal root ganglia and superior cervical ganglia, as well as some interneurons and dopaminergic neurons of the midbrain. At the behavioral level, ablation of Ca_V_2.2 channels results in increased locomotion, exploration, reduced startle [224,225], and reduced ethanol intake [226]. Ca_V_2.2 KO mice also show increased aggression and enhanced vigilance state related to disruption in random eye movement sleep [70]. All this combined suggests a role of Ca_V_2.2 channels in psychiatric disorders. 

Several studies have linked *CACNA1B* to SCZ, but also some *CACNA1B* risk variations are associated with BD and ASD. Purcell et al. identified a *de novo* variation (G/A) in patients with SCZ that introduces a premature stop codon in the proximal C-terminus of Ca_V_2.2 [195]. The intronic SNPs, rs7036881 and rs78178087, in *CACNA1B* have been found to be weakly associated with SCZ and the antipsychotic efficacy of paliperidone palmitate in a study with European patients [227]. In line with this, another study in a South African population found that the rs2229949 is linked to improved negative symptomatology during antipsychotic treatment [228]. Deletions in *CACNA1B* were detected in 16 patients and duplications of this same gene were detected in 10 patients with SCZ [229]. 

Several studies have reported that *CACNA1B* is linked to ASD, MDD, and BD. A monogenic duplication in *CACNA1B* has been linked to Asperger Syndrome, a condition that until recently was considered an ASD [230]. Pathway analysis of variations linked to ASD has shown that *CACNA1B,* together with *CACNA1C* and *CACNA1F*, converges on MAP kinase/cellular signaling and neuronal development/axon guidance [231]. *CACNA1B*, together with *CACNA1C* and *CACNA2D4*, has been also associated with suicide risk in patients with MDD [232]. Finally, WES of 200 individuals from 41 families identified 50 non-coding variations in *CACNA1B* that increase the risk for BD [201]. 

### 5.5. Ca_V_2.3

The *CACNA1E* gene encodes the Ca_V_2.3 channels. These channels are broadly expressed throughout the nervous system and are located in presynaptic terminals, dendritic spines, and some extrasynaptic sites [75]. Functional disruption of Ca_V_2.3 channels leads to increased anxiety-like behavior and impaired spatial memory [233,234]. Ca_V_2.3 deficient mice show reduced wake duration and increased slow-wave sleep, although these results depend on the strategy to knock out Ca_V_2.3 [235,236]. Nonetheless, this is relevant because alterations in sleep have been observed in patients with SCZ and BD.

Variations in the *CACNA1E* gene have been linked to ASD, MDD, SCZ, as well as some endophenotypes related to these conditions. In a study comprised of 209 families with no previous history of ASD, parent-child trios with sporadic autism and unaffected siblings were sequenced and a *de novo* variant in *CACNA1E* (G1209S) was identified in one patient [206]. G1209S is located in DIIIS3. A second *de novo* synonymous variation in *CACNA1E* located near a splice site and predicted to affect an exonic splicing regulator was identified in another patient with ASD [237]. In a genome-wide meta-analysis study of more than 135,000 cases with more than 340,000 controls, 44 significant risk loci for MDD were identified, including *CACNA1E* [238]. The SNP rs4652676 was linked to neuroticism and subjective well-being, which are endophenotypes associated with MDD [239]. The SNP rs704329 is implicated in the efficacy of serotonin reuptake inhibitors (SSRIs) in a Taiwanese population [240]. Similar to several other *CACNA1* genes, *CACNA1E* has been associated with SCZ as well as working memory related to cortex and cerebellum [158,241]. 

### 5.6. Ca_V_3.1

*CACNA1G* encodes the Ca_V_3.1 channel, a T-type channel member of the Ca_V_3 subfamily. Specific ablation of Ca_V_3.1 channels in the thalamus resulted in frequent and prolonged arousal, which reduced sleep [242,243]. Ca_V_3.1 channels also play a key role in prolonged unconsciousness by influencing thalamocortical rhythmicity [244]. Previous studies have identified risk variations of *CACNA1G* as genetic risk for ASD. A linkage study of sibling pairs with only male probands found a strong association of the chromosomic region 17q11–21, which contains among other genes, *CACNA1G* [245]. A later study confirmed *CACNA1G* as a novel candidate gene for ASD by identifying several SNPs within intron 9 with the strongest association relative to other genes present in the 17q11-21 region [246]. Alleles rs198538 and rs198545, together with some *CACNA1C* SNPs, were identified as risk variations for ASD [193]. Furthermore, a *de novo* synonymous variation in *CACNA1G* was identified in exome sequencing of 343 families with one proband and at least one unaffected sibling [207,247]. A *de novo* variation screening in childhood-onset cerebellar atrophy identified various disruptive variations in *CACNA1G*, some patients with this pathology exhibit autistic traits [248]. However new studies using transcriptome-wide association, which integrated GWAS with gene expression predictors from several databases from adult and fetal human brain, found no evidence of association between *CACNA1G* and ASD [249]. 

### 5.7. Ca_V_3.2

The *CACNA1H* gene encodes Ca_V_3.2 channels, also a T-type channel. This gene is normally associated with idiopathic epilepsy. However, multiple studies have found associations of *CACNA1H* with ASD and SCZ. Mice deficient in Ca_V_3.2 channels show increased anxiety-like behavior, impaired memory and reduced sensitivity to psychostimulants such as D-amphetamine and cocaine [250]. Ca_V_3.2 KO mice exhibit deficits in context-associated memories [251]. Ca_V_3.2 channels play a minor role in non-REM sleep [252,253]. 

Four missense variations (R212C, R902W, W962C and A1874V) were identified in a study of 461 probands with ASD and 480 ethnically matched individuals by targeted sequencing of the *CACNA1H* genomic region [254,255]. R212 is located DIS4, R902 in DIIS4, W962 in the P-loop between DIIS5 and DIIS6, and A1874 in the C-terminus. Functional analysis revealed that these variations produce loss of function of Ca_V_3.2 by reducing channel conductance, and/or shifting voltage-dependence of activation in the depolarizing direction [254]. However, these variations have low penetrance, and some of them were also found in unaffected individuals [254]. In a more recent study using ultra deep sequencing of 78 ASD candidate genes in the cerebellum and cortical samples of several ASD cases and neurotypical controls, a synonymous *CACNA1H* variation was found in the frontal cortex but not in cerebellum [256]. In this same study, a missense variation in the C-terminus (S1970C) was identified in a female diagnosed with ASD [256]. WES from more than 10,000 parents with only one child with ASD found *de novo* missense variations in *CACNA1H* [207,257]. Furthermore, a study of 262 ASD patients with their unaffected parents from Japan identified a disruptive *de novo* missense variations in *CACNA1H* (R1189C), which is located in the intercellular loop between DII and DII [258]. All of these studies support that *CACNA1H* is a susceptibility gene for ASD.

Two rare disruptive variations (7 bp and 2 bp deletions) in the DII-DIII linker in Ca_V_3.2 that are predicted to produce a frameshift were found in patients with SCZ [195]. Furthermore, a GWAS performed in a Swedish population, followed by a meta-analysis with previously identified genes associated with SCZ, found association of *CACNA1H* with this condition [259]. 

### 5.8. Ca_V_3.3

*CACNA1I* encodes Ca_V_3.3 channels, the third T-type channel member of the Ca_V_3 subfamily. Of the Ca_V_3 members, Ca_V_3.3 channels have the most depolarized threshold of activation, as well as the slowest opening and inactivation rate [27]. Ca_V_3.3 channels regulate sleep spindles. This is supported with evidence from animal models, mice with functional ablation of Ca_V_3.3 channels show impairment in sleep spindle generation [252,253,260,261]. Sleep spindles have been shown to be altered in patients with SCZ [184]. Not surprisingly, several studies have strongly linked *CACNA1I* to SCZ and related endophenotypes. Additional evidence also suggests risk variations of *CACNA1I* for ADHD and ASD.

Two rare, *de novo* missense variations of *CACNA1I* (R1346H and T797M) were identified by exome sequencing of trio samples that included 105 probands, parents, and unaffected siblings when available [262]. R797 and R1345 map to the P-loops of DII and DIII of Ca_V_3.3 respectively. Ca_V_3.3 was the only gene with more than one variation [262]. In particular, R1346H impairs N-glycosylation of Ca_V_3.3 channels preventing membrane targeting and thereby reducing overall calcium currents [263]. The functional consequences of T797M are unknown, however this mutant produces similar calcium currents relative to WT [263]. A study by the SCZ working group of the PGC validated *CACNA1I* as a risk gene for SCZ [175]. This claim has been supported in other GWAS. The intergenic SNPs between *RPS19BP1* and *CACNA1I*, rs5757717 and rs9611198, were found in a GWA study of an Ashkenazi Jewish population and an Irish population respectively [176,264]. The intronic SNP rs3788567 was identified with very high significance in an Ashkenazi Jewish population [176]. In a study of an Uyghur Chinese population that comprised 985 patients and 1218 neurotypical controls, six SNPs within *CACNA1I* were significantly associated with SCZ (rs132575, rs136805, rs713860, rs738168, rs5757760, rs575087) [265]. Furthermore, rs4522708, rs3788568, rs5750862 were significantly associated with SCZ in a Han Chinese population [266,267]. Interestingly, rs4522708 was also found in a study of a European population [175]. *CACNA1I* has been also associated with endophenotypes related to SCZ, such as cognitive ability and sleep spindle activity. A genome wide meta-analysis study identified an association of *CACNA1I* with cognitive ability [268]. The genomic region Chr22: 39975017:40016914, which spans across *CACNA1I* was associated with higher amplitude, longer duration and higher intensity of slow spindles in healthy adolescents [269].

A recent GWAS linked the rs199694726 in *CACNA1I* to impulsive behavior under extreme negative emotions [270]. Impulsive traits are a common endophenotype related to psychiatric disorders including ADHD [271]. Furthermore, a study containing 1013 probands of European descent at the Children’s Hospital of Philadelphia (CHOP) found a *CACNA1I* CNV (large deletion) associated with ADHD [272]. *CACNA1I* was also identified as a risk gene for ASD in the family-based association test [273]. The SNP rs5750860 was significantly associated with ASD in an another GWAS [193].

## 6. Genetic Associations between Auxiliary Subunit Genes *CACNA2D* (Ca_V_α_2_δ) and *CACNB* (Ca_V_β) and Psychiatric Disorders

In the previous section we summarized strong evidence linking several genes that encode Ca_V_α_1_ pore-forming subunits to psychiatric disorders. Given that the auxiliary subunits, Ca_V_α_2_δ and Ca_V_β, heavily influence membrane targeting and overall activity of Ca_V_α1, genes encoding these subunits are also strongly linked to psychiatric disorders. In this section, we will describe current studies associating the genes for Ca_V_α_2_δ and Ca_V_β with multiple psychiatric disorders. 

### 6.1. Ca_V_α_2_δ-1

The *CACNA2D1* gene encodes the Ca_V_α_2_δ-1 subunit. This subunit is highly expressed in skeletal muscle, the brain and peripheral nervous system [19], and some studies suggest that it is enriched in glutamatergic neurons [274]. No brain-related phenotypes have been reported for animal models with alterations in Ca_V_α_2_δ-1 expression (KO, knock in or overexpression) [275,276,277,278]. However, compensation by the Ca_V_α_2_δ subunits exists when one of them is disrupted [20].

Various genetic studies have implicated the *CACNA2D1* gene in psychiatric disorders including MDD, BD and SCZ. The genome-wide association metanalysis of MDD that identified *CACNA1E*, also found *CACNA2D1* as potentially druggable target for this condition [238]. Furthermore, in a genome-wide association environment study, a suggestive association was found for rs17156280 in *CACNA2D1* with an interaction between depressive state and stressful events [279]. A strong association with depressive traits including subjective well-being and neuroticism was found for the SNPs in *CACNA2D1*, rs258668 and rs258677 [239]. 

A metanalysis of data collected by the Bipolar Disorder Genome Study Consortium identified rs2367911 as a risk SNP for BD with comorbid binge eating. Indeed, networks/interactomes for *CACNA2D1* and apolipoprotein B gene (*APOB*) were the top two hits for BD and binge eating in this study [280]. The same study that identified risk variations for *CACNA1C* and other Ca_V_ genes in a Swedish population, found a disruptive variation in *CACNA2D1* that produces a frameshift associated with SCZ [195]. A study in a Japanese population a found CNV for *CACNA2D1* (a large deletion) in one patient with SCZ [116]. 

### 6.2. Ca_V_α_2_δ-2

*CACNA2D2* encodes Ca_V_α_2_δ-2. Although this protein is broadly expressed in the central nervous system, there is higher expression in the cerebellum relative to other areas of the brain, particularly in Purkinje cells [20]. In cortical tissue, some studies suggest that Ca_V_α_2_δ-2 is more abundant in interneurons than in glutamatergic neurons [274]. Deleterious effects from disrupting Ca_V_α_2_δ-2 have been observed in mouse genetic models including ataxia and seizures, however none of them are directly related to psychiatric disorders [8,281,282,283,284,285]. Purcell et al., found three *de novo* variations in *CACNA2D2* in patients with SCZ. Two of these three variations introduced premature stop codons, and the third one is predicted to disrupt a splice donor site [193]. A *CACNA2D2* variation (A900T) scored as a putative second hit in a study of 558 patients with SCZ in a Spanish population [286]. 

### 6.3. Ca_V_α_2_δ-3

*CACNA2D3* encodes Ca_V_α_2_δ-3. This protein was initially characterized as a target to treat pain, however recent studies suggest that the *CACNA2D3* is strongly linked to ASD, and to a lesser extent, SCZ and BD. Ca_V_α_2_δ-3 KO mice have alterations in pain processing at the central level [87], as well as enhanced cross-activation of brain regions involved in processing of auditory, olfactory and visual sensory information (cross-modal activation) [87,287,288]. Interestingly, patients with ASD and SCZ often exhibit altered pain perception [289,290] and synesthesia, the latter is a form of cross-modal activation [291,292]. 

The same WES that identified variations in *CACNA1G*, found another variation in *CACNA2D3* that is predicted to disrupt a splice junction associated with ASD (A/G) [247]. An inherited variation with splicing disruption was identified in a study of 2066 unique families with children diagnosed with ASD, the cohort consisted of 2618 children with ASD (1740 probands and 878 unaffected siblings) [293]. Furthermore, a *de novo* variation (E508Stop) predicting loss of function of Ca_V_α_2_δ-3 was found in two patients in an exome sequencing study that included 3871 ASD cases and 9937 ancestry controls. This study also identified several inherited variations in *CACNA2D3* which effect is unknown [212]. Analysis of CNVs in a study containing samples from 2478 families with children affected with ASD identified through the Simons Simplex Collection found association to a deletion in *CACNA2D3* [294]. In a study where 208 candidate genes were sequenced in 11,730 cases and 2867 controls, two *de novo* missense on *CACNA2D3* were identified (A773V and A275T) [295]. The SNP rs3773540 was among the top 15 SNPs contributing to ASD diagnosis as predicted by gene set enrichment analysis [194].

Previous studies have shown that the 3p14 genetic region is associated with SCZ and with an endophenotype related to the function of the temporal lobe, the antisaccade reflex. Interestingly, this genomic region contains *CACNA2D3* [296,297]. Pathway analysis of SNPs with significant risk for SCZ suggest association of *CACNA2D3* with the response to lurasidone, an antipsychotic used to treat SCZ [298]. Also, the genomic region 3p21.1_1 is enriched in for both SCZ and BD, this region contains *CACNA1D* and *CACNA2D3* among six different genes [299]. The SNP rs9849795 located in *CACNA2D3* is associated with functional brain connectivity inferred by functional magnetic resonance, this trait thought to be compromised in BD and SCZ, this study also identified association with SNPs in *CACNA1C*, *CACNA2D4* and *CACNB2* [300]. 

### 6.4. Ca_V_α_2_δ-4

*CACNA2D4* encodes the Ca_V_α_2_δ-4 subunit. This protein is abundantly expressed in the retina, but it is also found in pituitary and adrenal glands [88,301]. Despite the relatively low expression of Ca_V_α_2_δ-4 in the brain compared with other auxiliary subunits, several studies have identified the *CACNA2D4* as a risk gene for some psychiatric disorders. Although disruptive mutations in *CACNA2D4* in mice cause night blindness, as well as retinal degeneration, phenotypes related to the brain have not been reported [302,303,304].

The SNP rs1024582 located between *CACNA2D4* and *CACNA1C* was found highly significant in a cross-disorder study that included ADHD, BD, ASD, SCZ and MDD [158]. In a later study by Purcell et al., a *de novo* variation that produces a frameshift in *CACNA2D4* was identified in patients with SCZ [195]. The SNP rs4765847 was found to associate with DMN, an endophenotype of SCZ [300]. Furthermore, partial deletions of 35.7 kb in *CACNA2D4* was found in two unrelated patients with late onset BD I and one in control individuals [305]. These three deletions eliminate exons 17–26 in *CACNA2D4*, which comprise most of the Cache domain [305]. In a linkage disequilibrium study to detect SNP–SNP interactions that are common in complex diseases a single interaction between SNPs located near *RYR2* and *CACNA2D4* was found in samples of the Wellcome Trust Case Control Consortium (WTCCC) [306]. 

Genetic associations of *CACNA2D4* with MDD and ASD have been also identified. In a WES study in brain samples of suicide victims suffering from MDD and control subjects with MDD who died from other causes, a variation in a splice donor (C/A) in *CACNA2D4* was identified [232]. For ASD, a rare homozygous deletion was detected in a male proband that is predicted to affect *CACNA1C* and *CACNA2D4* (12p13.33) [307]. 

### 6.5. Ca_V_β_1_

*CACNB1* encodes Ca_V_β_1_. A splice variant of this subunit was originally identified in skeletal muscle (Ca_V_β_1a_) as the only partner of Ca_V_1.1, later it was demonstrated that splice variations of Ca_V_β_1_ are also expressed in the brain (Ca_V_β_1b_, Ca_V_β_1c_, and Ca_V_β_1d_), particularly in cerebral cortex, habenula, hippocampus and olfactory bulb [45]. Null mice for Ca_V_β_1_ subunit show reduced muscle mass and die of asphyxiation after birth, heterozygous are relatively normal and no phenotypes linked to higher order brain functions have been reported [308]. 

Some studies suggest association of the *CACNB1* with ASD, BD and SCZ; however, the evidence is scarce. A metanalysis of five genome-wide linkage scans in 634 affected sibling pairs found a suggestive association between the chromosome region 17p11.2–q12 and ASD, this region comprises *CACNB1*, however this finding requires further replication [309]. For BD, increased *CACNB1* expression was reported in IPSCs derived from patients with BD relative to IPSCs from their unaffected relatives [310]. In this study, *CACNA1G* and *CACNA1E* were downregulated [310]. Regarding SCZ, only one GWAS has linked *CACNB1,* together with other calcium channel genes, with SCZ and working memory across multiple ages in healthy individuals [241]. 

### 6.6. Ca_V_β_2_

*CACNB2* encodes the Ca_V_β_2_ subunit. Ca_V_β_2_ is widely expressed in the brain, heart, and other tissues such as lung, liver and pancreas. *CACNB2* has the largest number of splice variants among the *CACNB* genes, all these splice variants are abundant in the heart and brain [45]. *CACNB2,* together with *CACNA1C,* is one of most consistently found risk genes for psychiatric disorders, particularly SCZ and BD. Evidence of association of *CACNB2* with MDD and ASD has also been reported. In animal models, Ca_V_β_2_ KO mice lack cardiac contractions during development, therefore, are embryonically lethal [311]. Heart-specific rescue of Ca_V_β_2_ KO mice resulted in survival and several phenotypes such as deafness and blindness related to impairments of inner ear and retina respectively, but not the brain [312,313]. It remains to be determined the behavioral role of Ca_V_β_2_ expressed exclusively in the brain. 

Several SNPs in *CACNB2* have been linked to SCZ with high significance including rs7893279, rs7099380, rs17691888, rs2799573, and rs10508558. The allele rs7893279 was identified in a Psychiatric Genomics Consortium study for SCZ [175], rs7099380 in an Ashkenazi Jew population [176], rs17691888 in a Swedish population and was further confirmed using a regulatory trait concordance approach to prioritize SNPs and genes within SCZ loci [314,315], rs2799573 has been identified across multiple disorders including SCZ, BD, ADHD and ASD [158,316]. rs10508558 was identified in genome-wide metanalysis for SCZ [317]. Other SNPs in *CACNB2* such as rs17661538 have been linked to antipsychotic responses of clozapine [318], and rs1277738 has been found across multiple disorders and also linked to DMN [300]. Other intronic SNPs in *CACNB2* are also associated with working memory and brain activity [241]. 

Similar to *CACNA1C*, SNPs in *CACNB2* have shown strong association with BD and other psychiatric disorders. In fact, the some of the SNPs in *CACNB2* that are associated with BD are also associated with SCZ. For example, an association of rs11013860 with BD and SCZ was identified in a Han Chinese and Taiwanese populations [192]. A study using a pleiotropy-informed conditional false discovery rate, which improved detection of common variations associated with BD, identified rs7083127 [319]. *CACNB2* has also been associated with binge eating and BD in a second study [280]. rs2489198, rs4747340, rs7083127, rs12247369, rs2799573 have been linked to the five major disorders ADHD, SCZ, ASD, BD, and MDD [158]. Furthermore, several SNPs in *CACNB2* are linked to the response to SSRIs [320]. 

In a WES study, three variations in *CACNB2* were found in ASD probands but not in controls (G167S, S197F, and F240L), although with incomplete segregation. All three variations affect the kinetics of inactivation of calcium currents [321]. In a second study that included 85 family quartets (two parents and two affected siblings), the variations V2D was identified, but the functional effect of this variant is unknown [322].

### 6.7. Ca_V_β_3_

*CACNB3* encodes the Ca_V_β_3_ subunit and is mostly expressed in the brain and to some extent in heart, aorta, and kidney. Ca_V_β_3_ KO mice have enhanced hippocampus-dependent learning and memory which correlates with increased long-term potentiation in excitatory hippocampal synapses [323]. These mice also show defects in working memory, reduced anxiety-like behavior and increased aggression [324]. This suggest that the Ca_V_β_3_ subunit has important behavioral implications on phenotypes like working memory, which is linked to several psychiatric disorders including SCZ and BD. Previous studies have shown associations of the *CACNB3* with these conditions. The SNPs rs2070615 and rs11168751 were found to confer risk to BD in a European population [163,325]. QLTs in *CACNB3* have also been linked to both BD and ADHD [326]. Finally, pathway analysis has confirmed associations of *CACNB3* with SCZ [327]. 

### 6.8. Ca_V_β_4_

*CACNB4* encodes the Ca_V_β_4_ subunit, and together with Ca_V_β3, is one of the most abundant Ca_V_β subunits in the brain. Ca_V_β_4_ subunit is the most commonly found Ca_V_β in complex with Ca_V_2 channels suggesting an important role of this subunit in presynaptic transmitter release [328]. Naturally occurring Ca_V_β_4_ KO mice, also known as lethargic, exhibit several phenotypes including ataxia, seizures, absence epilepsy, and paroxysmal dyskinesia [5,329]; however, none of them are linked to endophenotypes associated with psychiatric disorders. Several studies have linked *CACNB4* to MDD, anxiety disorders and SCZ [259,330]. 

## 7. Ca_V_ Modulators for the Treatment of Psychiatric Disorders

Given the large amount of evidence from multiple studies implicating Ca_V_ genes in the pathophysiology of psychiatric disorders, it is worthwhile to consider targeting Ca_V_α_1_, Ca_V_α_2_δ, and Ca_V_β subunits as a potential therapeutic strategy to treat these disorders. Although several drugs targeting Ca_V_α_1_, Ca_V_α_2_δ subunits already exist, they are typically prescribed to treat cardiovascular conditions, pain, and epilepsy [14]. Nimodipine, isradipine, verapamil and diltiaziem target Ca_V_1 channels and are currently prescribed to treat cardiovascular conditions, but now are being explored to treat psychiatric disorders (Table 2). Drugs with anti-epileptic and analgesic effects such as gabapentin and pregabalin are now being explored as a novel approach to treat anxiety [331]. Similarly, topiramate, a drug that has several targets including Ca_V_2.2 and Ca_V_2.3 channels, has shown some promise to treat posttraumatic stress disorder (PTSD) comorbid with alcohol dependence [332]. Ca_V_2.2 channel blockers such as Z160 and CNV2197944 are being considered to treat anxiety (Table 2) [14]. Finally, lamotrigine, a drug that blocks Ca_V_2.3 channels, is used to treat BD and MDD [333,334].

Currently, several trials targeting Ca_V_s and auxiliary subunits have been completed or are being performed (www.clinicaltrials.gov) (Table 2). The L-type channel blockers, nimodipine and isradipine are being evaluated for their effects on cognitive performance in patients with SCZ. Ethosuximide, a drug that blocks Ca_V_3 channels, is being tested for treatment-resistant depression. Gabapentin is also being tested for bipolar disorder. Various clinical trials have been completed testing the efficacy of lamotrigine (a drug that targets Ca_V_2.3 channels [335,336]) in BD as well as clinical trials for MDD and SCZ. Drugs targeting Ca_V_s that are showing promise in animal models are the Ca_V_3 channel enhancer, Sak3. This drug has been found to reduce depressive-like behaviors in mice by increasing serotonin and dopamine levels [337,338].

## 8. Conclusions

Modern analysis of large cohorts has shed a tremendous amount of light on the genetic risks associated with psychiatric disorders. Techniques such as next generation sequencing, microarrays, linkage studies, endophenotype analysis and computer modeling are increasing our chances to elucidate the cellular and molecular mechanisms underlying psychiatric disorders. Although most genetic studies strongly suggest that multiple genes are associated with psychiatric disorders, risk variations in Ca_V_ gene have been consistently associated with the five major psychiatric disorders SCZ, MDD, ADHD, ASD, and BD. This nicely aligns with the neuronal functions of Ca_V_ genes.

The Ca_V_α_1_, Ca_V_α_2_δ, and Ca_V_β subunits are relatively well-known pharmacological targets. Several studies have demonstrated their involvement in neuronal firing, axon guidance, neuronal development, synapse formation and activity-dependent function. However, a major challenge is to link risk variations of Ca_V_ genes to their pathophysiological functions in the context of psychiatric disorders. Studies on the SNP rs1006737 in *CACNA1C* are leading the way on this— several studies been performed at the molecular, cellular and behavioral level to elucidate how this risk variation is involved in BD. However, in addition to individual risk variations, it is important to weigh the compounded effect of individual variations as they interact with other genes, and with the environment. Machine learning is becoming a powerful means to integrate information arising from genetic studies to elucidate the various mechanisms that are likely to underlie psychiatric disorders, as shown with PsychEncode.

For therapeutic purposes, tissue expression of Ca_V_ genes should be taken into account. For example, Ca_V_1.2 channels are promising targets for BD and SCZ; however, their robust expression in the heart and blood vessels poses a challenge for intervention. Further studies should aim at blocking or activating specific Ca_V_s present in the brain but not in the heart. Alternative splicing is a possible path for drug specificity, because *CACNA1C* splice variants in the heart are substantially different from the ones in the brain [31]. Nonetheless, Ca_V_s offer an intriguing viable option to develop novel treatments for psychiatric disorders.

## Figures and Tables

**Figure 1 ijms-20-03537-f001:**
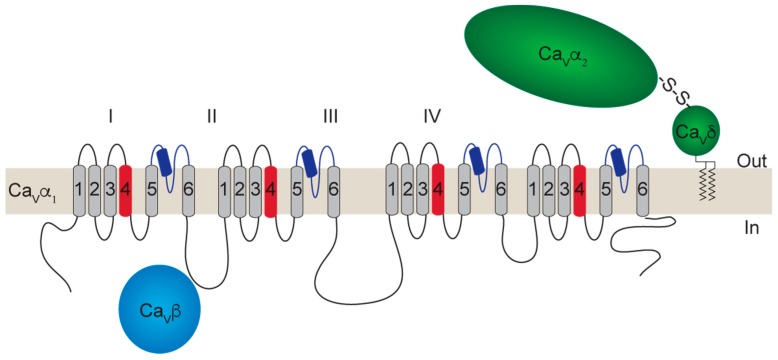
Schematic representation of a voltage-gated calcium channel (Ca_V_) complex. The Ca_V_α_1_, Ca_V_α_2_δ, and Ca_V_β subunits are depicted. Transmembrane segments of the Ca_V_α_1_ subunit (S1–6) are shown arranged in the four domains (DI-IV), the voltage sensors are indicated in red (S4), and the reentrant loop between S5 and S6 (P-loop) in dark blue. The glycosylphosphatidylinositol (GPI)-anchored Ca_V_α_2_δ is shown in green, and the cytoplasmic Ca_V_β subunit in light blue.

**Table 1 ijms-20-03537-t001:** Nomenclature and classification of voltage-gated calcium channels Ca_V_α_1_ subunits based on sequence similarity and biophysical properties.

Protein Name *Gene Name* Current Type	Expression Profile	Subfamily	Threshold of Activation
Ca_v_1.1 *CACNA1S* L-type	Skeletal muscle (myocytes)	Ca_V_1	HVA (associated with CaVα2δ and Cavβ subunits)
Ca_v_1.2 *CACNA1C* L-type	Brain, cardiovascular system (smooth muscle of blood vessels, sinoatrial and atrioventricular nodes, cardiomyocytes), pancreatic islets, adrenal medulla (chromaffin cells), intestinal and bladder smooth muscle, sympathetic and sensory ganglia, pituitary gland.		
Ca_v_1.3 *CACNA1D* L-type	Brain, cochlear and vestibular hair cells, retina, heart (sinoatrial and atrioventricular nodes, cardiomyocytes), pancreatic islets, adrenal medulla (chromaffin cells) and adrenal cortex, sympathetic and sensory ganglia, pituitary gland.		
Ca_v_1.4 *CACNA1F* L-type	Retina (photoreceptors)		
Ca_v_2.1 *CACNA1A* P/Q-type	Brain (broadly expressed but dominant in cerebellar Purkinje cells and glutamatergic neurons), spinal cord motor neurons, sympathetic and sensory ganglia, pancreas and pituitary	Ca_V_2	
Ca_v_2.2 *CACNA1B* N-type	Brain (broadly expressed but dominant in monoaminergic neurons, as well as cholecystokinin expressing interneurons), sympathetic and sensory ganglia		
Ca_v_2.3 *CACNA1E* R-type	Brain, heart (atrial myocytes), testis, pituitary, pancreatic islets, gastrointestinal system, lungs		
Ca_v_3.1 *CACNA1G* T-type	Brain, heart (sinoatrial node), aorta, immune system (T-cells), bone, lung, glands (pancreas, ovary, testis)	Ca_V_3	LVA (associated with *CACHD1*)
Ca_v_3.2 *CACNA1H* T-type	Brain, heart (sinoatrial node), kidney, liver, adrenal cortex, smooth muscle, sensory ganglia (low threshold mechanoreceptors)		
Ca_v_3.3 *CACNA1I* T-type	Brain, thyroid, spleen, small intestine, adrenal gland		

**Table 2 ijms-20-03537-t002:** Summary of genetic links between Ca_V_ genes and psychiatric disorders, classical Ca_V_s, inhibitors and potential pharmacological strategies to treat psychiatric disorders using drugs that target Ca_V_s (drugs in clinical trials). — indicates that agents targeting this channel with potential use to treat psychiatric disorders are yet to be identified.

Ca_V_	Associated Disorder	Pharmacological Inhibitors	Potential Therapeutic Intervention for Psychiatric Fisorders
Ca_V_1.1	—	Dihydropyridines	Nimodipine (SCZ) Isradipine (BD, SCZ) Verapamil (BD) Diltiazem (BD)
Ca_V_1.2	ASD, SCZ, BD, MDD, ADHD		
Ca_V_1.3	ASD, SCZ, BD, MDD, ADHD		
Ca_V_1.4	—		
Ca_V_2.1	SCZ, ADHD, MDD	ω-Agatoxin IVA	—
Ca_V_2.2	SCZ, ASD, MDD	ω-Conotoxin GVIA	CNV2197944 (anxiety) Z160 (anxiety)
Ca_V_2.3	ASD, MDD, SCZ	SNX 482	Topiramate (PTSD)
Ca_V_3.1	ASD	TTA-A2, TTA-P2, ProTx-I, ProTx-II	Sak3 (MDD) Ethosuximide (MDD)
Ca_V_3.2	ASD, SCZ		
Ca_V_3.3	SCZ, ADHD, ASD		
Ca_V_α2δ-1	MDD, BD, SCZ	Gabapentin, pregabalin	Pregabalin (anxiety, SCZ) Gabapentin (anxiety, mood disorders)
Ca_V_α2δ-2	SCZ		
Ca_V_α2δ-3	ASD, SCZ, BD	—	—
Ca_V_α2δ-4	ASD, SCZ, BD, MDD, ADHD		
Ca_V_β_1_	ASD, BD, SCZ	—	—
Ca_V_β_2_	ASD, SCZ, BD, MDD, ADHD		
Ca_V_β_3_	ASD, BD, SCZ		
Ca_V_β_4_	MDD, SCZ, anxiety disorders		

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
