# Peer review of "Genetic Associations between Voltage-Gated Calcium Channels and Psychiatric Disorders"

_ijms, 2019, doi:10.3390/ijms20143537_

Round 1

Reviewer 1 Report

ID: ijms-540927

The study by Andrade et al., entitled “Genetic associations between voltage-gated calcium channels and psychiatric disorders” reviews the implications of voltage-gated Ca2+ channels in human psychiatric conditions, particularly bipolar disorders, schizophrenia, autism, anxiety, depression and attention disorders. Ca2+ channels are described as major players in neuronal communication, and which mutations generate channelopathies. The review briefly describes the structure and pharmacology of Cav channels. The main part of the review is devoted to the links between gene mutations (mainly in the α subunit) and psychiatric disorders, revealed by epidemiological and/or electrophysiological/pharmacological studies.

Major

This review is interesting because many articles look at the relationships between Cav channels and neurological diseases, but few explore psychiatric deficits. I think substantial changes could improve the overall level of this review. As it stands, it cannot be published. Authors should improve the overall presentation and make the reading of this journal more accessible.

In general, the review could be better structured: the template for preparing the articles in IJMS provide for cutting the article into section, subsection, subsubsection (https://www.mdpi.com/journal/ijms/instructions). It is necessary in the review of Andrade et al. to review this point. Section 4 for example could be advantageously divided.

Table 1 is obviously crucial in this review. But I think it is absolutely necessary to complete it by adding a column indicating in which brain/peripheral structure (cerebellum, muscle ...) or to which activity (release of glutamate, GABA) Cav channels are associated. This is indicated in the text (section 3) but it would be very complementary to make it appear in Table 1. This type of information appears very clearly in FIG. 2 from Zamponi et al., 2015. Pharmacol Rev. 67: 821-870.

Many models of KO mice for Cav channels have been created and show different phenotype (pain, heart rate, neurological diseases). The authors do not address this aspect for transgenic models in the case of psychiatric diseases: it must be indicated if there are some (and which) or not (no animal model for these specifically human deficits without possible transposition between man and the animal).

Section 5 5 (Overview of databases and resources to study the links between genetic risk variations and psychiatric disorders) is unnecessary in this review because it is a consortium work that can be cited, but which goes beyond the subject of this review. This section can be reduced to one or two sentences and included in the other sections.

In general, this review deals with human channelopathies associated with pyschiatric deficits. These channelopathies originate from mutations with gain/loss of function, at the level of genes encoding Cav channels. However, it would be easier for the reader to name the channels by their protein names (Cav1.2, Cav1.3) than by the name of their gene (CACNA1C, CACNA1D). I therefore propose that the authors make these modifications both in the body of the text and in the figures, with the exception of Table 1. The sentence (L377-378) "Although risk variations in CACNA1S and CACNA1F have been identified in GWAS and WES studies for BD and SCZ” would become " although risk variations in Cav1.1 and Cav1.4 have been identified in GWAS and WES studies for BD and SCZ ".

Table 2 indicates "Potential therapeutic intervention for psychiatric disorders"; but when we look at the molecules that are proposed, we find drugs with FDA-approval for certain non-psychiatric uses (verapamil for hypertension or arrhythmia) and molecules with approval for a psychiatric indication (lamotrigine for BD and related depression). So, this table has potential or authorized therapeutic uses? Thank you for clarifying this point.

Minor

Instructions for Authors state that “All Figures, Schemes and Tables should be inserted into the main text close to their first citation and must be numbered following their number of appearance”. Figure1 and Tables 1 & 2 should be inserted within the text.

L29: The calcium that enters through Cavs is crucial…

L31: Some Cavs are multi-protein complexes… : please precise (“some” is not clear).

Pharmacological antagonists of Cav are summarily described. For instance, “Cav2.1 is sensitive to w-agatoxin IVA” is really incomplete. Please refer to Lewis et al., 2012. Pharmacol Rev: Conus Venom Peptide Pharmacology. Or precise somewhere that the pharmacological aspects of Cav are scarcely described, especially since this review focuses, among other things, on pharmacological tools in the context of psychiatric disorders.

L51: This section should be renamed “Structure and pharmacology of voltage-gated calcium channels”

L63: and Cav2.3 to the SNX-482 peptide toxin

L271: Genetic associations between Cav channels and psychiatric disorders

Change subsection titles by Cav names. It will facilitate the reading.

L289: the SNP rs1006737 allele

L322-323: Is it “rs2007044 is also associated with increased concentrations of glutamate and or

glutamine in subcortical regions” ? Precise.

L371: SH3 and Multiple

L413: Variations

Figure 1: please numerate S1 to S6 in each transmembrane domain. It would be better, for your review, to insert the residues and domains concerned by point mutations or stop codon linked to psychiatric diseases.

L637: Currently several clinical trials

L642-643: [278,279]) in BD as well as clinical trials for MDD and SCZ.

L681: DI-IV (not D1-IV)

L689-1108: no spacing between 2 references

L1109-1547: line space between two references

Table 2: Replace the name of the genes (CACNA1C) by the name of the proteins (Cav1.2). What is TBD ?

Table 3: Please replace “classical modulators” by “pharmacological inhibitors”. Remove the bullet points (useless). Justify.

Reviewer 2 Report

This is a nice review of the involvement of voltage-gated calcium channels in psychiatric disorders. Increasing evidence linked this important group of channels to psychiatric disorders, namely schizophrenia, bipolar disorder, and autism spectrum disorders, etc. The authors provided an exhaustive review of the literature and analysis of the state of the field. Additionally, the review also evaluated the therapeutic potential by targeting on voltage-gated Ca2+ channels. Overall, this is a well-written update of the field that should have a broad interest by the readers from basic science researchers to clinicians.

Round 2

Reviewer 1 Report

The authors have made substantial efforts to respond to all the comments I sent them and their review is significantly improved. It seems quite publishable now. I leave it to editors and proofreaders to correct minor errors in the draft.